# Psychometric Properties of the Italian Version of the Toileting Habit Profile Questionnaire-Revised (THPQ-R) in Children with Autism Spectrum Disorder

**DOI:** 10.3390/children10091528

**Published:** 2023-09-08

**Authors:** Anna Berardi, Giovanni Galeoto, Martina Ruffini, Rachele Simeon, Jerónimo González-Bernal, Jesús Angel Seco-Calvo

**Affiliations:** 1Department of Human Neurosciences, Sapienza University of Rome, 00185 Rome, Italy; anna.berardi@uniroma1.it; 2IRCSS Neuromed, Via Atinense, 18, 86077 Pozzilli, Italy; 3Institute of Biomedicine (IBIOMED), Universidad de León, 24071 León, Spain; jesus.seco@unileon.es; 4IRCCS San Raffele Pisana, 00163 Rome, Italy; martina.ruffini@uniroma1.it; 5Department of Anatomical, Histological, Forensic and Orthopaedic Sciences, Sapienza University of Rome, 00161 Rome, Italy; rachele.simeon@yahoo.com; 6Health Sciences, University of Burgos, 09001 Burgos, Spain

**Keywords:** sensory integration, occupational therapy, autism spectrum disorder, assessment

## Abstract

Introduction: Children with autism spectrum disorder (ASD) often seem not to feel the need to go to the bathroom in whatever context they are in and may suffer from hypo-responsiveness. Recent studies show a correlation between sensory problems, constipation, and fecal incontinence in children. This requires an assessment tool to identify evacuation issues in children with sensory dysfunction. Therefore, the purpose of this study is to validate the Italian version of The Toileting Habit Profile Questionnaire-Revised (THPQ-R) in children with ASD. Methods: The inclusion criteria were a confirmed diagnosis of ASD and an age between 3 and 8 years old. The Toileting Habit Profile Questionnaire Revised (THPQ-R) was recently validated in the Italian language on a healthy population. The Italian version has 17 items with two alternative response options, frequently/always or never/rarely, based on how often the behavior occurs. The THPQ-R questionnaire is easy to administer, interpret, and manage. For concurrent validity, we used the Italian version of the Sensory Processing Measure and Sensory Processing Measure—Preschool. Results: For the THPQ-R validation, 43 participants were recruited. The internal consistency, reporting an α value of 0.763, showed that the THPQ-R was a very reliable scale. The THPQ-R appears to be correlated only with the domains “Social Participation T-point”, “Hearing T-point”, and “Planning and Ideas T-point”. However, these correlations are not statistically significant. **Conclusion**: The THPQ-R highlights positive results regarding validity and reliability and shows a positive correlation between defecation disorders and social participation.

## 1. Introduction 

The inability to independently manage personal hygiene and sphincter control significantly affects the quality of life of children and their caregivers.

In particular, autonomy in hygiene and independence in evacuation, followed by the acquisition of voluntary bowel continence are two of the most important goals for childhood and should occur from ages 3 to 6 years. The inability to achieve these goals can limit the child’s independence and social participation. Sphincter control disorders, particularly constipation, are very common in children [1,2]. More specifically, studies have shown that children with autism spectrum disorder (ASD) have a high prevalence of gastrointestinal disorders, most frequently constipation [3,4,5,6,7,8]. In addition to the characteristic symptomatology of individuals with ASD, namely impaired social interactions, impaired and atypical verbal and nonverbal communication, and repetitive and unusual behaviors and games [9], there is also the presence of gastrointestinal disorders, and specifically, constipation is also related to behavioral disorders related to food selectivity [10]. The prevalence of ASD in children in the general population is estimated to range from 0.6 to 1% [11]. Despite the numerous available studies, a retrospective study defined 8.5% of children with ASD to be diagnosed with retentive fecal incontinence (RFI) according to the old Rome II criteria [12]. Fecal incontinence, especially RFI [13,14], is a very common disorder in children. In most cases, it is not taken into account and is therefore undertreated. As a more disabling consequence, it causes a reduction in social participation, resulting in slower development from a social relations point of view. The term RFI is generally used for specifying the occurrence of constipation. The determinant cause of RFI is unclear; despite this, a child who develops RFI has an identified pattern of behavior that determines the development and persistence of the condition.

In most cases, a child with RFI has a habit of holding stool because of pain; fecal retention may also be determined by the individual’s fear or refusal to sit on the toilet. In addition, it was also noted in one study that some of the children, despite a lack of motor or psychological disabilities, had an overreaction to the bodily sensations and stimuli felt in the daily act of evacuation. This aspect can be attributed to the association that exists between the presence of sensory deficits and the consequent reduction in the individual’s autonomy and well-being.

It was therefore postulated that impairment in the elaboration and integration of sensory feedback may contribute to the development of some atypical behaviors.

Children with ASD often seem not to feel the need to go to the bathroom in whatever context they are in and may suffer from hypo-responsiveness [15]. The encoding of bodily signals is impaired, and the child often finds himself/herself wet or dirty, thus experiencing great discomfort. 

Problems at this level may manifest themselves as failing to feel the need to perform needs or a failure to perceive the exact sensation of signals from the bladder and/or intestinal tract.

On the other hand, the voluntary withholding of feces or the tendency to avoid evacuation finds its place in the child’s strategy for avoiding the sensation of expulsion, which is often felt as painful. One can find, therefore, an explanation if one hypothesizes interceptive hyper-responsiveness [16], experiencing the sensation more abnormally than how other children perceive it. Over-exaggerated responses to ordinary sensory signals and excessive sensory reactivity have been linked to atypical habits related to certain activities, such as self-care. 

Recent studies have confirmed the connection between sensory problems, fecal incontinence, and constipation in children [17,18]. This requires an assessment tool to identify evacuation issues in children with sensory dysfunction. The outcome measures currently used to assess sensory integration dysfunction are the Sensory Processing Measure and the Sensory Processing Measure—Preschool (SPM-P) [19]. Dr. Isabelle Beaudry designed and validated The Toileting Habit Profile Questionnaire-Revised (THPQ-R) [20,21,22] in Spain to fill this gap. Deficits from a sensory integration point of view are linked to two different aspects and include poor perception/hypo-reactivity or hyperreactivity. In the first case, the subject does not autonomously recognize the stimulus of evacuation, but only feels it at the pressing moment when this occurs. In this case, the child does not have the material time to go to the toilet and consequently may not notice that he/she is dirty. 

In the case of hyperreactivity, however, the child refuses to defecate in the bathroom because for him/her, defecation is an extremely unpleasant sensation, and refuses to use the toilet or is afraid of the latter. These two main dimensions of sensory integration and defecation are considered in the THPQ-R. In the original validation study, the discriminant and concurrent validity of the THPQ-R were examined. As for the discriminative validity, the original validation study considered children with or without constipation. The study showed that a low THPQ-R hyperreactivity score corresponded to the absence of functional constipation (FC). When the THPQ-R hyperreactivity score was above average, the child presented FC. The difference in the mean THPQ-R hyperreactivity score between children with FC and those without FC was statistically significant. The Virginia Encopresis-Constipation Apperception Test (VECAT-S) defecation scale was used for concurrent validity. Pearson’s index demonstrated a strong correlation between the hyperreactivity section of both scales’ scores. Furthermore, in a recent study (2019), the correlation between hyperreactivity and functional defecation disorders (FDDs) was demonstrated. A persistent and high frequency of behaviors described in the THPQ-R corresponded to a high level of sensory hyperreactivity, i.e., a low Short Sensory Profile score corresponded to a low THPQ-R score. 

The THPQ-R was translated into Italian and validated in 2022 on a population of healthy subjects [23], and it reported good psychometric properties in Spanish and English translations. Given the correlation between defecation disorders, sensory integration, and ASD, this study aims to validate the Italian version of THPQ-R in the ASD population.

## 2. Materials and Methods

### 2.1. Subject Enrollment

The study was performed per the ethical standards in the 1964 Declaration of Helsinki and its later amendments. The participants were enrolled at IRCCS San Raffaele Pisana in a pediatric outpatient clinic from May 2022 to September 2022. All the patients gave informed consent to participate in this study. The inclusion criteria were a confirmed diagnosis of ASD and an age between 3 and 8 years old. 

### 2.2. Instruments

The Toileting Habit Profile Questionnaire Revised (THPQ-R) [23] was recently validated in the Italian language on a healthy population. The Italian version has 17 items with two alternative response options: frequently/always or never/rarely, based on how often the behavior occurs. The THPQ-R questionnaire is easy to administer, interpret, and manage.

In addition, the same caregiver completes the “cover letter,” consisting of 9 questions, which provide basic information regarding the child. The exploratory factor analysis (EFA) performed on the Italian version of the THPQ-R extracted 17 items. A total of 64% of the variance distinguished the items into two main domains: sensory hyperreactivity and sensory hypo-reactivity. The first domain included 15 items (sensory hyperreactivity), while the second domain included 2 items (sensory hypo-reactivity). The internal consistency was 0.76 (Cronbach’s alpha value) for the first domain and 0.75 for the second domain, demonstrating good stability across the items. The removal of some items resulted in a decrease in the alpha value and, as a consequence, internal consistency also decreased. All items thus proved to be significant within the questionnaire. Cross-cultural validity showed that 5-year-olds had lower scores on the THPQ-R. This shows a greater defecation problem [23].

The Sensory Processing Measure (SPM) and the Sensory Processing Measure—Preschool (SPM-P) are a set of assessment protocols to measure sensory processing difficulties, praxis, and social participation in school- and preschool-aged children [19,24].

The preschool version (SPM-P) assessed children aged 2 to 5, while the school version (SPM) assessed children aged 6 to 12.

The instruments are composed of two different protocols: Home and School.

-Home Protocol: composed of 75 items and filled out by the parent or caregiver of the child, with an internal consistency of 0.921 for the total sensory system.-School Protocol: composed of 62 items and filled out by the child’s main teacher, with an internal consistency of 0.957 for the total sensory system.

Each protocol provided eight normative reference standard scores: Social Participation (SOC); Vision (VIS); Hearing (UDI); Touch (TAT); Body Awareness (COR); Balance and Movement (EQU); Planning and Ideas (IDE); and Total Raw Score (TOT).

SOC and IDE were not included within the TOT.

The standard score for each scale allowed the child’s functioning to be classified into one of three interpretive ranges: typical, some difficulty, or definite dysfunction.

### 2.3. Procedures

The administration was carried out within the Hospitalization and Care Institutes of a Scientific Character (IRCCS) San Raffaele Pisana in Rome. Patients were recruited through medical records. All participants meeting the inclusion criteria were contacted by telephone or email to complete a form either in paper format or in a digital format (online) via Google Forms. Each parent or tutor of the eligible participants was advised of the aims and procedures of the present investigation, and those interested in participating were given a file containing the informed consent, personal form, and THPQ-R; at the same time, they also compiled the House Protocol belonging to the SPM and the SPM-P.

### 2.4. Statistical Analysis

In the following validation study, the Consensus-Based Standards for the Selection of Health Status Measurement Instrument (COSMIN) checklist was followed, evaluating reliability and validity through internal consistency and construct validity [25]. We also used the exploratory factor analysis and Pearson’s Correlation Coefficient measurement to determine the association between the THPQ-R and the Italian versions of the SPM. The correlation coefficients <0.30, <0.60, and ≥0.60 were considered “poor”, “moderate”, and “strong” correlations, respectively [26]. Through the calculation of Cronbach’s alpha value, internal consistency was analyzed. The alpha values of 0.7, 0.8, and 0.9 represented a fair, good, and excellent degree of internal consistency, respectively. For the statistical analysis, an Excel sheet was created to report the statistical data obtained from the administration of the THPQ-R and SPM. An IBM^®^ SPSS^®^ tool was used to carry out the statistical analysis.

## 3. Results

Table 1 shows the demographic characteristics of the population. For the THPQ-R validation, 43 participants were recruited, with a mean age of 6 years old, and 79.1% were male. Of the participants, 9.3% were 3 years old, 11.6% were 4 years old, 20.9% were five years old, 25.6% were 6 years old, and finally, 16.3% were 7 and 8 years old. As shown in Table 2, the exploratory factor analysis (EFA) was performed to calculate the construct validity of the THPQ-R in the ASD population. The exploratory factor analysis (EFA) was performed to calculate the construct validity of the THPQ-R in the ASD population. The result of the Kaiser–Meyer–Olkin sample adequacy test (KMO) was 0.62, which meant a medium sample size. Bartlett’s test of sphericity was statistically significant (*p* < 0.001), so the matrix correlations were high enough to be non-zero. The EFA extracted 17 items that explained 54% of the variance, contained in four main factors. Factor 1 included six items (1,2,3,4,5,14), Factor 2 included six items (6,8,9,13,15,16), Factor 3 included four items (7,11,12,17), and finally Factor 4 included one item (10).

### 3.1. Reliability

To check the homogeneity among the items, Cronbach’s alpha value was calculated to investigate internal consistency. The rating scale had a strong internal consistency, reporting an α value of 0.763, showing that the THPQ-R results had very good internal consistency, that is, very good interrelation among the items. In addition, as can be seen from Table 3, all items were important for the scale and had a good relation between each other. The alpha values if an item was deleted ranged from 0.8 to 0.76, even though if we eliminated items 10 and 15, the internal consistency value would slightly increase. However, these two items are considered fundamental according to previous studies. 

### 3.2. Validity

Twenty-two participants, in addition to the THPQ-R, also filled out the Sensory Processing Measure (SPM) (ages 6 to 12) and the preschool version (SPM-P) (ages 3 to 5) so that the construct validity of the THPQ-R could be calculated. There was a possible correlation between the scores of the THPQ-R, the SPM-P, and the SPM, shown in Table 3, as determined through the calculation of Pearson’s Correlation Coefficient. As shown in Table 4, the THPQ-R appeared to be correlated with Pearson’s Correlation Coefficient between 0.285 (Vision) and 0.060 (Body Awareness) for the SPM and between 0.156 (Touch) and 0.632 (Planning and Ideas) for the SPM-P. The strongest correlations were shown with Social Participation (SPM-P), Hearing (SPM-P) and Planning and Ideas (SPM-P); however, these correlations were not statistically significant (*p* > 0.05).

## 4. Discussion

In daily clinical practice, health care professionals usually do not consider sensory integration disorders as possible factors that influence and increase defecation difficulties in children with ASD. On the contrary, recent studies have shown how these disorders decisively affect toilet use behavior in this population^15^. Because of the above, the use of screening in clinical practice that identifies these types of disorders is essential. 

The purpose of the present study was to assess evacuation problems related to sensory dysfunction by validating the Toileting Habit Profile Questionnaire-Revised (THPQ-R) rating scale on a sample of children with autism spectrum disorder (ASD) to make sure it is valid and reliable.

The internal consistency was assessed, calculating the value of Cronbach’s alpha, which was found to be equal to 0.763. This value is consistent with the Italian version of the tool analyzed on a sample of Italian children with typical development (0.719). 

To analyze the concurrent validity of the THPQ-R, 22 participants also completed the Sensory Processing Measure (SPM) and Sensory Processing Measure—Preschool (SPM-P). The association between the responses given by the subjects was measured with Pearson’s Correlation Coefficient. This is the first study analyzing the concurrent validity of the scale. The THPQ-R appeared to be correlated with the domains “Social Participation T-point” (SPM-P), “Hearing T-point” (SPM-P), and “Planning and Ideas T-point” (SPM-P). However, these correlations were not statistically significant (*p* > 0.05). Unfortunately, the results were not statistically significant because the administration of the SPM-P and SPM was carried out on only a few subjects (SPM-P-n 7 and SPM-n 15). Convergent analysis showed that children between the ages of 3 and 5 years with ASD had a statistically significant correlation in the domains of “Hearing T-point” (SPM-P), “Social Participation T-point” (SPM- P), and “Planning and Ideas T-point” (SPM-P).

Children aged 3–5 years are characterized by the inability to acquire voluntary bowel continence and dependence on hygiene, which may limit their social participation, making them experience great distress [27]. In addition, from the literature, it was possible to observe that children with autism spectrum disorder who exhibit irritable bowel syndrome and abdominal pain syndrome in this age group have a stronger auditory reflex (hyperexcitation) [28]. Previous studies have assessed concurrent validity through the Virginia Encopresis-Constipation Apperception Test (VECAT-S) [29] questionnaire. Unfortunately, the same outcome measure could not be used in this study because this questionnaire has not been validated in or culturally adapted to Italy. Despite this, the previous study showed a strong correlation between the score of the THPQ-R and VECAT-S, and thus, a strong correlation between hyper/hypo-reactivity and defecation problems. This study, on the other hand, showed a positive correlation between defecation disorders and social participation; therefore, a higher THPQ-R score corresponded to reduced social participation and deficit in planning and organizing. Finally, by comparing the mean scores obtained with healthy Italian children and Italian children with ASD, it is possible to affirm that typically developing children have a higher score than children with ASD. This highlights, therefore, that children with ASD show greater difficulty in evacuation, linked to sensory dysfunction. The results define how 5-year-old children have more defecation-related difficulties, compared with other-age children. Scores in the THPQ-R were lower in both groups for this age range. Defecation-related problems, including functional constipation, can be common in children in this age group. Major causes include the child’s dietary behaviors or may be associated with episodes related to daily life such as the start of school or parental toilet training. In Italy, elementary school starts at 5 years of age and is an important change and beginning in the child’s life. This could explain the presence of more defecation problems in this age group; however, this hypothesis is not supported by other studies.

Limitations of the study: The small number of participants may have influenced some of the results, e.g., the non-significant correlation coefficients. It would be beneficial to investigate responsiveness (i.e., the standardized response mean, minimal clinically important difference, and clinically important difference) in a larger sample of patients.

Conclusion: In conclusion, from the data emerging in the present study, it is possible to understand that the sample with ASD had an internal consistency comparable to or slightly higher than the sample with typical development. Furthermore, given that the study had a positive outcome in terms of internal consistency, it is possible to state that a screening tool was developed that is capable of correctly evaluating the presence of deficiencies in evacuation in children with ASD. It is hoped that this project will create interest for further studies in this field of investigation and give rise to cooperation between occupational therapists and pediatricians or gastroenterologists in the treatment of toilet disorders in children with ASD.

This study demonstrates how defecation problems are an issue that occupational therapists are obliged to consider during assessment in children with ASD. Occupational therapists should also set up treatment based on this aspect since it is a sphere that affects autonomy and social participation. The instrument proved to be brief, schematic, and easy to understand and administer, representing a useful tool both in research and especially in clinical practice. Prior to this validation, there was no instrument that could accurately assess the presence of defecation problems related to sensory integration disorders in children with autism; to date, researchers strongly suggest the use of this instrument in daily clinical practice.

## Figures and Tables

**Table 1 children-10-01528-t001:** Demographic characteristics of the included participants.

	Frequency	Percentage
Age		
3	4	9.3
4	5	11.6
5	9	20.9
6	11	25.6
7	7	16.3
8	7	16.3
Gender		
Male	34	79.1
Female	9	20.9

**Table 2 children-10-01528-t002:** Exploratory factor analysis (EFA) results.

	1 Communication	2 Sensitivity	3 Habits	4 Sensory
1 Mio figlio si nasconde per fare la cacca	0.431			
2 Mio figlio mi chiede un pannolino quando ha bisogno di fare la cacca	0.736			
3 Mio figlio preferisce farsi la cacca addosso nonostante sia vicino al vasino o al water	0.837			
4 Mio figlio rifiuta di sedersi sul vasino o sul water per fare la cacca, però accetta di sedersi per fare la pipì	0.684			
5 Mio figlio rifiuta o non si sente a suo agio sedendosi sul water o sul vasino sia per fare pipì che per fare la cacca, anche se a casa sua.	0.629			
6 Mio figlio trattiene la cacca o l’impulso di farla.		0.863		
7 Mio figlio ha delle abitudini poco comuni per fare la cacca che implicano azioni o luoghi che non sono normalmente associati con il fare la cacca.			0.753	
8 Mio figlio sembra provare dolore quando fa la cacca, anche se la cacca ha una consistenza morbida.		0.421		
9 Mio figlio si rifiuta di fare la cacca in posti che non siano casa sua.		0.770		
10 Mio figlio ha delle reazioni di disgusto per l’odore della sua cacca.				0.764
11 Mio figlio rifiuta di pulirsi o di essere pulito dopo aver fatto la cacca.			0.400	
12 Mio figlio mostra di aver paura o si rifiuta di fare certe cose che normalmente si fanno in bagno come tirare lo sciacquone.			0.342	
13 Mio figlio ha bisogno di distrarsi con qualcosa mentre fa la cacca (con libri, giochi); questo sembra aiutarlo ad accettare di fare la cacca.		0.326		
14 Mio figlio mostra di essere sensibile al gusto o consistenza degli elementi ricchi di fibre o di farmaci che aiutano a fare la cacca.	0.423			
15 Mio figlio ha iniziato a sentire l’impulso di fare la cacca da molto piccolo (prima dei 12 mesi). Quando mio figlio si lamentava in un certo modo veniva messo sul vasino per fare la cacca.		0.397		
16 Mio figlio non sembra sentire lo stimolo di fare la cacca.		0.324		
17 Mio figlio non si rende conto che si è sporcato (di cacca) i vestiti o non prova fastidio ad essere sporco.			0.724	

**Table 3 children-10-01528-t003:** Internal consistency of the Toileting Habit Profile Questionnaire Revised (THPQ-R) based on Cronbach’s alpha analysis.

	Mean	Standard Deviation	Cronbach’s Alpha If Item Deleted
My child hides to poop.	1.67	0.474	0.762
My child asks for a diaper when he feels the need to poop.	1.77	0.427	0.769
My child prefers to poop in his clothing although the potty or toilet is nearby.	1.67	0.474	0.775
My child refuses to sit on the potty or the toilet to poop, but will accept to pee in the potty or toilet.	1.79	0.412	0.772
My child refuses or seems uncomfortable sitting on the toilet or potty for both peeing and pooping, even at home.	1.79	0.412	0.769
My child withholds poop or resists the urge to poop.	1.58	0.499	0.762
My child follows an unusual ritual when pooping which involves actions or places not typically associated with pooping or with the age of the child.	1.67	0.474	0.772
My child seems to feel pain when pooping, even if the poop is soft.	1.74	0.441	0.771
My child refuses to poop outside of the home.	1.70	0.465	0.769
My child shows exaggerated disgust at the smell of his poop.	1.95	0.213	0.802
My child refuses to wipe or be wiped after pooping.	1.86	0.351	0.782
My child shows fear or refusal related to certain features of the bathroom, such as fear of flushing the toilet.	1.86	0.351	0.780
My child needs to pay attention to something else while pooping (a book, a game); this seems to help him/her tolerate the sensation of pooping.	1.63	0.489	0.780
My child is sensitive to taste and/or food textures making it difficult to accept laxative medicine or high fibre foods.	1.53	0.505	0.772
My child started to feel the urge to poop from a very young age (before 12 months). When my child complained in a certain way he was put on the potty to poop.	1.86	0.351	0.804
Total Cronbach’s Alpha			0.763

**Table 4 children-10-01528-t004:** Construct validity of the Toileting Habit Profile Questionnaire Revised (THPQ-R) based on Pearson’s Correlation Coefficient with the Sensory Processing Measure (SPM) and the preschool version (SPM-P).

	Total Score THPQ-R
Point T Social Participation (SPM)	0.232
Point T Vision (SPM)	0.281
Point T Hearing (SPM)	0.285
Point T Touch (SPM)	0.122
Point T Body Awareness (SPM)	0.060
Point T Balance and Movement (SPM)	0.099
Point T Planning and Ideas (SPM)	0.159
Point T Social Participation (SPM-P)	0.509
Point T Vision (SPM-P)	0.221
Point T Hearing (SPM-P)	0.514
Point T Touch (SPM-P)	0.156
Point T Body Awareness (SPM-P)	0.214
Point T Balance and Movement (SPM-P)	0.156
Point T Planning and Ideas (SPM-P)	0.632

## Data Availability

Data that support the findings of this study are available from the corresponding author upon reasonable request.

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
