# Peer review of "Psychometric Properties of the Italian Version of the Toileting Habit Profile Questionnaire-Revised (THPQ-R) in Children with Autism Spectrum Disorder"

_children, 2023, doi:10.3390/children10091528_

Round 1
Reviewer 1 Report
The aim of the paper is to present results from a validation study on the Toileting Habit Profile Questionnaire-Revised (THPQ-R) among children with ASD in Italian context. Two factors were extracted explaining 64% of the variance. Associations with additional scales were also studied in order to explore the concurrent validity of the THPQ-R. My suggestion to present in the Methods section examples of the items of the scales used in the analysis and the response options. The authors should add also a section on the limitations of the study. The small number if the participants may include some of the results, i.e. the non-significant correlation coefficients.
Typos and misspelled words need to be edited.
Reviewer 2 Report
Thank you for the opportunity to review this study entitled “Psychometric properties of the Italian version of the Toileting Habit Profile Questionnaire-Revised (THPQ-R) in children with Autism Spectrum Disorders” (children-2566180).
The paper present the psychometric properties of the Italian THPQ-R involving a sample of children with diagnosis of Autism Spectrum Disorders.
In my opinion, the research topic is relevant, and the study is interesting. Parallelly, some issues need to be addressed before the paper will be suitable for publication.
· Abstract: please, avoid references in the abstract.
· Measures: please include information about the internal consistency of the used scales.
· Procedures: more information should be provided.
· Data analysis: CFA should be replicated in this sample and the fit indices should be shown.
· Discussion: a clear section showing the limitations and the suggestion for future research should be added at the end of the discussion.
· Conclusion should further highlight the clinical implication of this research.
Best wishes
